# ALWAYS-SPARSE TRAINING WITH GUIDED STOCHASTIC EXPLORATION

## ABSTRACT

The excessive computational requirements of modern artificial neural networks (ANNs) are posing limitations on the machines that can run them. Sparsification of ANNs is often motivated by time, memory and energy savings only during model inference, yielding no benefits during training. A growing body of work is now focusing on providing the benefits of model sparsification also during training. While these methods greatly improve the training efficiency, the training algorithms yielding the most accurate models still materialize the dense weights, or compute dense gradients during training. We propose an efficient, always-sparse training algorithm which improves the accuracy over previous methods. Additionally, our method has excellent scaling to larger and sparser models, supported by its linear time complexity with respect to the model width during training and inference. We evaluate our method on CIFAR-10/100 and ImageNet using ResNet, VGG, and ViT models, and compare it against a range of sparsification methods.

## 1 INTRODUCTION

Artificial neural networks (ANNs) are currently the most prominent machine learning method because of their unparalleled performance on a broad range of applications, including computer vision (Voulodimos et al., 2018; O'Mahony et al., 2019), natural language processing (Young et al., 2018; Otter et al., 2020), and reinforcement learning (Arulkumaran et al., 2017; Schrittwieser et al., 2020), among many others (Liu et al., 2017; Wang et al., 2020; Zhou et al., 2020). To improve their representational powers, ANNs keep increasing in size (Neyshabur et al., 2019; Kaplan et al., 2020). Recent large language models, for example, have widths surpassing 10,000 units and total parameter counts over 100 billion (Brown et al., 2020; Rae et al., 2021; Chowdhery et al., 2022). However, with their increasing size, the memory and computational requirements to train and make inferences with these models becomes a limiting factor (Hwang, 2018; Ahmed & Wahed, 2020).

A large body of work has addressed problems arising from the immense model sizes (Reed, 1993; Gale et al., 2019; Blalock et al., 2020; Hoefler et al., 2021). Many studies look into sparsifying the model weights based on the observation that the weight distribution of a dense model is often concentrated around zero at the end of training, indicating that the majority of weights contribute little to the function being computed (Han et al., 2015). By utilizing sparse matrix representations, the model size and the number of floating-point operations (FLOPs) can be reduced dramatically. Moreover, previous work has found that for a fixed memory or parameter budget, larger sparse models outperform smaller dense models (Zhu & Gupta, 2017; Kalchbrenner et al., 2018).

Biological neural networks have also evolved to utilize sparsity, in the human brain only one in about 10 million possible connections is realized (Gerum et al., 2020). The sparsity of the brain is seen as an important property for learning efficiency (Watts & Strogatz, 1998; Bullmore & Sporns, 2009; Pessoa, 2014). To this end, sparse ANNs have shown better generalization properties (Frankle & Carbin, 2018; Chen et al., 2021), suggesting that sparsity is indeed an effective regularizer.

Early works in ANN sparsity removed connections, a process called *pruning*, of a trained dense model based on the magnitude of the weights (Janowsky, 1989; Ström, 1997), resulting in a more efficient model for inference. While later works improved upon this technique (Guo et al., 2016; Dong et al., 2017; Yu et al., 2018), they all require at least the cost of training a dense model, yielding no efficiency benefits during training. Consequently, the resulting sparse models are limited in size by the largest dense model that can be trained on a given machine.

In light of the aforementioned limitation, the Lottery Ticket Hypothesis (LTH), surprisingly, hypothesized that there exists a subnetwork within a dense over-parameterized model that when trained with the same initial weights will result in a sparse model with comparable accuracy to that of the dense model (Frankle & Carbin, 2018). However, the proposed method for finding a *winning ticket* within a dense model is very computationally intensive, as it requires training the dense model (typically multiple times) to obtain the subnetwork. Moreover, later work weakened the hypothesis for larger ANNs (Frankle et al., 2019). Despite this, it was still an important catalyst for new methods that aim to find the winning ticket more efficiently.

Efficient methods for finding a winning ticket can be categorized as: pruning before training and dynamic sparse training (DST). The before-training methods prune connections from a randomly initialized *dense* model (Lee et al., 2018; Tanaka et al., 2020). This means that they still require a machine capable of representing and computing with the dense model. In contrast, the DST methods start with a randomly initialized *sparse* model and change the connections dynamically throughout training, maintaining the overall sparsity (Mocanu et al., 2018; Mostafa & Wang, 2019; Evci et al., 2020). In practice, DST methods achieve higher accuracy than the pruning before training methods (see Section 4.3). The first DST method, Sparse Evolutionary Training (SET), periodically prunes the connections with the lowest weight magnitude and *grows* new connections uniformly at random (Mocanu et al., 2018). RigL improves upon the accuracy of SET by growing the connections with the largest gradient magnitude instead (Evci et al., 2020). These connections are expected to get large weight magnitudes as a result of gradient descent. While RigL achieves good accuracy, it has an important limitation: it requires periodically computing the *dense* gradients.

We present a DST algorithm that is *always sparse* and improves upon the accuracy of previous methods using a simple, yet highly effective, method that we call Guided Stochastic Exploration (GSE). In short, when changing the connections, our method first efficiently samples a subset of the inactive connections for exploration. It is then guided by growing the connections with the largest gradient magnitude from the sampled subset. The key insight is that the accuracy compared to RigL is maintained, in fact even improved, when the size of the subset is on the same order as the number of active connections, that is, the weight sparsity and gradient sparsity are similar. We additionally show analytically that, when using the common Erdős–Rényi model initialization, our method improves the training time complexity of RigL from $O(n^2)$ to $O(n)$ with respect to the model width $n$. We evaluate the accuracy of our method experimentally on CIFAR-10/100 and ImageNet using ResNet, VGG, and ViT models and compare it against a range of sparsification methods at 90%, 95%, and 98% sparsity, indicating the percentage of zero weights. Our method promises to enable the training of much larger and sparser models.

## 2 RELATED WORK

A wide variety of methods have been proposed that aim to reduce the size of ANNs, such as dimensionality reduction of the model weights (Jaderberg et al., 2014; Novikov et al., 2015), and weight quantization (Gupta et al., 2015; Mishra et al., 2018). However, this section only covers model sparsification methods as they are the most related to our work. Following Wang et al. (2019), the sparsification methods are categorized as: pruning after training, pruning during training, pruning before training, and dynamic sparse training.

**After training** The first pruning algorithms operated on dense trained models, they pruned the connections with the smallest weight magnitude (Janowsky, 1989; Thimm & Fiesler, 1995; Ström, 1997; Han et al., 2015). This method was later generalized to first-order (Mozer & Smolensky, 1988; Karnin, 1990; Molchanov et al., 2019a;b) and second-order (LeCun et al., 1989; Hassibi & Stork, 1992; Frantar et al., 2021) Taylor polynomials of the loss with respect to the weights. These methods can be interpreted as calculating an importance score for each connection based on how its removal will effect the loss (Guo et al., 2016; Dong et al., 2017; Yu et al., 2018).

**During training** Gradual pruning increases the sparsity of a model during training till the desired sparsity is reached (Zhu & Gupta, 2017; Liu et al., 2021). Kingma et al. (2015) introduced variational dropout which adapts the dropout rate of each unit during training, Molchanov et al. (2017) showed that pruning the units with the highest dropout rate is an effective way to sparsify a model. Louizos et al. (2018) propose a method based on the reparameterization trick that allows to directly

optimize the $L^0$ norm, which penalizes the number of non-zero weights. Alternatively, DeepHoyer is a differentiable regularizer with the same minima as the $L^0$ norm (Yang et al., 2019).

**Before training** The Lottery Ticket Hypothesis (LTH) (Frankle & Carbin, 2018; Frankle et al., 2019) started a line of work that aims to find a sparse model by pruning a dense model before training (Liu et al., 2018). SNIP uses the sensitivity of each connection to the loss as the importance score of a connection (Lee et al., 2018). GraSP optimizes gradient flow to accelerate training (Wang et al., 2019), however, Lubana & Dick (2020) argue that it is better to preserve gradient flow instead. Tanaka et al. (2020) highlight a problem in the aforementioned methods: they suffer from layer collapse in high sparsity regimes, that is, during the pruning phase all the connections of a layer can be removed, making the model untrainable. They propose SynFlow, which prevents layer collapse by calculating how each connection contributes to the flow of information using a path regularizer (Neyshabur et al., 2015), similar to Lee et al. (2019).

**Dynamic sparse training** The methods in this last category, including ours, start with a randomly initialized sparse model and change the connections dynamically throughout training while maintaining the same sparsity. This involves periodically pruning a fraction of active connections and growing the same number of inactive connections. SET was the first method and used magnitude pruning and random growing of connections (Mocanu et al., 2018). DeepR assigns a fixed sign to each connection at initialization and prunes those connections whose sign would change during training (Bellec et al., 2018). DSR prunes connections with the lowest global weight magnitude and uses random growing, allowing the connections to be redistributed over the layers and used in layers where they contribute the most (Mostafa & Wang, 2019). SNFS improves upon the previous methods by using an informed growing criteria: it grows the connections with the largest gradient momentum magnitude (Dettmers & Zettlemoyer, 2019). RigL makes this more efficient by only periodically calculating the dense gradients (Evci et al., 2020). However, calculating the dense gradients is still a limitation for training very large and sparse models. Top-KAST addresses this limitation by sparsely updating the dense model using only the connections with the largest weight magnitude (Jayakumar et al., 2020). While Top-KAST uses sparse gradients, it requires maintaining and updating the dense model from which the sparse model is periodically reconstructed. Moreover, we will show in Section 4.3 that Top-KAST achieves lower accuracy at higher sparsities.

## 3 METHOD

Our main contribution is an efficient dynamic sparse training algorithm which is always sparse, that is, at no point is the dense model materialized, all forward passes use sparse weights, and it exclusively calculates sparse gradients. The sparsity is maintained even when changing the connections because our method only computes the gradients for a subset of the inactive connections. This subset is randomly sampled during each round of growing and pruning. The connections within the subset with the largest gradient magnitudes are then selected to grow. We call our method Guided Stochastic Exploration (GSE), referring to the stochastic sampling process of the subset and the selection of the connections by largest gradient magnitude, which provides guidance during the exploration. Although we are not the first to present an always-sparse training method, for example, SET also achieves this, among these methods, ours obtains the highest accuracy—equal or higher than RigL.

Figure 1: Illustrated set sizes. The solid gray area becomes the active set in the next step. The set $\mathbb{W}$ is dotted to indicate that it is never materialized.

We will use the following notation and definitions. With $\mathbb{W}$ we denote the set of all possible connections, which is never materialized; $\mathbb{A} \subseteq \mathbb{W}$ is the set of non-zero or active connections whose weights are optimized with gradient descent; $\mathbb{S} \subseteq (\mathbb{W} \setminus \mathbb{A})$ is a subset of inactive connections; $\mathbb{G} \subseteq \mathbb{S}$ is the set of grown connections; and $\mathbb{P} \subseteq \mathbb{A}$ is the set of pruned connections. Figure 1 illustrates the relations between the various connection sets. Note that we abuse the notation of the connection sets to be per layer or global interchangeably.

GSE is designed to incorporate the strengths of both SET and RigL, the two baseline methods can be regarded as opposing extremes in their exploration strategy. On one end, SET conducts unbiased exploration, enabling it to be highly efficient. On the opposite end, RigL employs a greedy

strategy by selecting connections which appear to get the largest weight magnitudes. The exploration dynamics of GSE are, in turn, determined by the interaction of the subset and the grow set. When the subset is much larger than the grow set, in the limit including all possible connections, then the exploration is nearly greedy since the sampling process has little influence on which connections are selected. On the other hand, when the sets are equal in size, then the grown connections are fully determined by the subset sampling process. The exploration strategy of GSE is thus a hybrid of exploration guided by the selection of the largest gradient magnitude.

In the remainder of this section, we will discuss which probability distributions were considered for sampling the subset and how to efficiently sample from them in the case of fully-connected layers. In Appendix E, we discuss how our method can be applied to convolutional layers simply by interpreting them as batched fully-connected layers. We conclude this section with our complete dynamic sparse training algorithm and the time complexity of each step.

## 3.1 SUBSET SAMPLING

A connection $(a, b) \in \mathbb{W}$ is fully specified by the input and output units it connects. Thus, to ensure an efficient sampling process, we can sample connections at random by independently sampling from distributions over the input and output units of a layer, which induces a joint probability distribution over all the connections in that layer. With this process the probability of sampling a connection is never explicitly represented, just the probability of sampling each unit is represented which requires only linear, instead of quadratic, memory and $O(n)$ time to draw $O(n)$ samples using the alias method (Vose, 1991). Formally, connections are sampled from the discrete probability distributions over the input and output units of a layer $l$: $a_i \sim f^{[l]}$ and $b_i \sim g^{[l]}$. Since this process could sample duplicate or active connections, the set difference $\mathbb{S} \leftarrow \mathbb{S} \setminus \mathbb{A}$ with the active set is computed in $O(n)$ time using hash tables, resulting in a subset of connections whose size is bounded from above by the number of samples drawn, containing only inactive connections.

To sample the subset $\mathbb{S}$, perhaps the simplest distribution is uniform over the set of all possible connections $\mathbb{W}$. This reflects an unbiased exploration of the connections. However, the distributions f and g do not have to be uniform, it might instead be useful to bias the subset towards connections that will contribute more at decreasing the loss, that is, those that have a large gradient magnitude. To this end, we investigate two other distributions. The first distribution is called GraBo, its probabilities are proportional to an upper bound of the gradient magnitude, given as follows:

$$ f^{[l]} := \frac{\left|\boldsymbol{h}^{[l-1]}\right|\mathbf{1}}{\mathbf{1}^\top\left|\boldsymbol{h}^{[l-1]}\right|\mathbf{1}}, \quad g^{[l]} := \frac{\left|\boldsymbol{\delta}^{[l]}\right|\mathbf{1}}{\mathbf{1}^\top\left|\boldsymbol{\delta}^{[l]}\right|\mathbf{1}} $$

where $\mathbf{1}$ is an all ones vector, and $\boldsymbol{h}^{[l]}$ and $\boldsymbol{\delta}^{[l]}$ are the activations and gradients of the loss at the output units of the $l$-th layer, respectively. The second distribution is called GraEst, its probabilities are proportional to an unbiased estimate of the gradient magnitude, given as follows:

$$ f^{[l]} := \frac{\left|\boldsymbol{h}^{[l-1]}\boldsymbol{s}\right|}{\mathbf{1}^\top\left|\boldsymbol{h}^{[l-1]}\boldsymbol{s}\right|}, \quad g^{[l]} := \frac{\left|\boldsymbol{\delta}^{[l]}\boldsymbol{s}\right|}{\mathbf{1}^\top\left|\boldsymbol{\delta}^{[l]}\boldsymbol{s}\right|} $$

where $\boldsymbol{s}$ is a vector of random signs. To compute the probabilities, both distributions can reuse the computation of the activations and the gradients during the forward and backward pass, respectively. A more detailed discussion of the distributions is provided in Appendix A. In Section 4.2, we assess which of these probability distributions is the most appropriate for training sparse models.

## 3.2 DYNAMIC SPARSE TRAINING

Our complete dynamic sparse training algorithm uses a similar model initialization procedure as SET and RigL (Mocanu et al., 2018; Evci et al., 2020), each layer is initialized as a random bipartite graph using the Erdős–Rényi random graph generation algorithm (Erdős & Rényi, 1959). The number of active connections $|\mathbb{A}|$ of a layer $l$ is given by $\lceil \epsilon(n^{[l-1]} + n^{[l]}) \rceil$ where $n^{[l]}$ are the number of units in the $l$-th layer and $\epsilon \in \mathbb{R}^+$ is a parameter that controls the sparsity of the model. This initialization ensures that the number of active connections in a layer is proportional to its width, that is, $|\mathbb{A}| = O(n)$. Moreover, Evci et al. (2020) showed that the Erdős–Rényi initialization also achieves better accuracy than uniform sparsity assignment. We verify their findings in Appendix F.

The weights are initialized using the procedure proposed by Evci et al. (2019) which adapts standard weight initialization to take into account the actual connectivity of each unit and scales the weight distribution for each unit accordingly.

---

**Algorithm 1** Efficiently growing and pruning connections

---

**Input:** training step $t$, prune-grow schedule $\alpha, T_{\text{end}}$, probability distributions f and g, subset sampling factor $\gamma$, set of active connections $\mathbb{A}$ of size $O(n)$ and their weights $\theta$.

1: $\mathbb{S} \leftarrow \text{sample\_connections}(f, g, \lceil \gamma|\mathbb{A}| \rceil) \setminus \mathbb{A}$           ▷ $O(n)$
2: $\alpha_t \leftarrow \text{cosine\_decay}(t; \alpha, T_{\text{end}})$           ▷ $O(1)$
3: $k \leftarrow \min(\lceil \alpha_t|\mathbb{A}| \rceil, |\mathbb{S}|)$           ▷ $O(1)$
4: $\mathbb{G} \leftarrow \text{topk}(|\text{grad}(\mathbb{S})|, k)$           ▷ $O(n)$
5: $\mathbb{P} \leftarrow \text{topk}(-|\theta|, k)$           ▷ $O(n)$
6: $\mathbb{A} \leftarrow (\mathbb{A} \setminus \mathbb{P}) \cup \mathbb{G}$           ▷ $O(n)$
7: $\theta \leftarrow \text{update\_weights}(\theta, \mathbb{A}, \mathbb{P}, \mathbb{G})$           ▷ $O(n)$

---

Once the random sparse model is initialized, the weights of the active connections are optimized using stochastic gradient descent (SGD). Every $T$ training steps new connections are grown and the same number of connections are pruned. We provide pseudocode for our pruning and growing procedure in Algorithm 1, which operates on a single layer for simplicity. The maximum fraction of pruned active connections is cosine annealed from $\alpha \in (0, 1)$ at the start of training to zero at $T_{\text{end}}$. This ensures that there is more exploration in the early stages of training.

To determine the set of connections to grow $\mathbb{G}$, we start by evaluating one of the probability distributions described in Section 3.1. Then, $\lceil \gamma|\mathbb{A}| \rceil$ connections are sampled from the input and output distributions, where $\gamma \in \mathbb{R}^+$ is a hyperparameter that adjusts the number of samples. The subset $\mathbb{S}$ contains all randomly sampled connections that are not in the active set $\mathbb{A}$. Now that the subset of inactive connections is determined, their gradient magnitude is computed. The grow set contains the largest $k$ gradient magnitude connections from the subset, where $k$ is the minimum between $\lceil \alpha_t|\mathbb{A}| \rceil$ and $|\mathbb{S}|$ to ensure that the grow set is not larger than the subset. The $k$ largest elements can be found in $O(n)$ time using the introspective selection algorithm (Musser, 1997).

The pruned set $\mathbb{P}$, in turn, contains the $k$ active connections with the smallest weight magnitude, also determined in $O(n)$ time. Although magnitude pruning has been observed to cause layer collapse among pruning before training methods, Tanaka et al. (2020) investigated layer collapse and found that the training process is implicitly regularizing the weight magnitudes which minimizes the risk of layer collapse for DST methods like ours. Lastly, the active set and the weights are updated to reflect the changed connections. Pseudocode for the entire training process is provided in Appendix G. It is important to stress that the dense model is not materialized at any point, only the active connections are represented, and all the operations are always sparse.

## 4 EXPERIMENTS

At this point, in order to verify our claims, we need to determine what the minimum number of subset samples is that maintains the maximum accuracy. We answer this question in Section 4.2 and then compare the accuracy of our method against other sparsification methods in Section 4.3. We present accuracy results for ImageNet in Section 4.4. In Section 4.5, we explore the relation between model scale and accuracy for a fixed number of active connections by increasing the sparsity of larger models. Lastly, in Section 4.6, we compare the FLOPs of GSE with those of RigL.

### 4.1 EXPERIMENT SETUP

We evaluate our method and compare it to baselines on the CIFAR-10 and CIFAR-100 datasets (Krizhevsky et al., 2009), in addition to the ILSVRC-2012 ImageNet dataset (Deng et al., 2009). To establish how each method compares across model architectures we experiment with two convolutional neural networks (CNNs), ResNet-56 (He et al., 2016) and a 4 times downscaled version of VGG-16 (Simonyan & Zisserman, 2014), in addition to ViT (Dosovitskiy et al., 2020) modified for small datasets (Beyer et al., 2022) such that all models have roughly the same number of parameters. The bias and normalization parameters are kept dense since they only contribute

marginally to the size and cost of a model. Similar to Wang et al. (2019), we evaluate all methods at 90%, 95%, and 98% sparsity. The high levels of sparsity emphasize the differences between the methods. In addition, in Appendix L, we show that sparse matrix formats for unstructured sparsity become more efficient than dense matrix formats from 90% sparsity, further motivating the need for such high levels of sparsity. We repeat each experiment three times with random initializations and report the mean and plot the 95th percentile. We adopt the implementation of the baselines based on their available code.

All experiments use the same optimization settings in order to isolate the differences in sparsification. Similar to Evci et al. (2020) and Lubana & Dick (2020), we use SGD with a momentum coefficient of 0.9, an $L^2$ regularization coefficient of 0.0001, and an initial learning rate of 0.1 which is dropped by a factor of 10 at epochs 80 and 120. The ViT experiments have one exception, they use an initial learning rate of 0.01. We use a batch size of 128 and train for 200 epochs. We also apply standard data augmentation to the training data, described in Appendix D. Running all the reported experiments took on the order of 2000 GPU-hours with a single NVIDIA Tesla V100 16GB GPU per training run on an internal cluster.

The LTH procedure described by Frankle & Carbin (2018), denoted as Lottery in the results, uses iterative pruning with iterations at 0%, 50%, 75%, 90%, 95% and 98% sparsity. The gradual pruning method by Zhu & Gupta (2017), denoted as Gradual in the results, reaches the target sparsity at the second learning rate drop and prunes connections every 1000 steps. Following Tanaka et al. (2020), the pruning before training methods use a number of training samples equal to ten times the number of classes to prune the dense model to the target sparsity. Except for SynFlow, which uses 100 iterations with an exponential decaying sparsity schedule to prune the dense model. All the DST methods use the same update schedule: the connections are updated every 1000 steps, and similar to Evci et al. (2020), the fraction of pruned active connections $\alpha$ is cosine annealed from 0.2 to 0.0 at the second learning rate drop. This is because Liu et al. (2021) showed that DST methods struggle to recover from pruning when the learning rate is low. The pruning and growing procedures are applied globally for all methods, maintaining the overall sparsity. This approach generally outperforms local, layer-wise sparsity (Mostafa & Wang, 2019).

## 4.2 NUMBER OF SUBSET SAMPLES

First, we want to determine how the size of the subset affects the accuracy. In addition, we are interested in comparing the accuracy obtained by the three distributions: uniform, gradient upper bound (GraBo), and gradient estimate (GraEst). The number of subset samples (bounding the subset size) is set proportional to the number of active connections $|\mathbb{A}|$, with $|\mathbb{S}| \leq \lceil \gamma |\mathbb{A}| \rceil$ and $\gamma \in \{0.25, 0.5, 1, 1.5, 2\}$. The results are shown in the first three columns of Figure 2 and include the baseline accuracy of RigL for comparison.

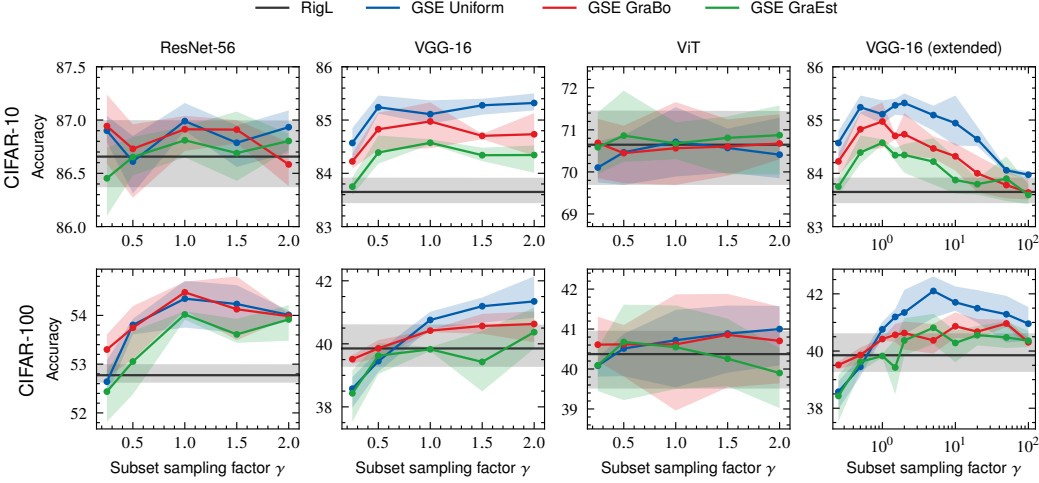

Figure 2: Accuracy of each distribution while increasing the number of subset samples (bounding the size of the subset) compared against RigL at 98% sparsity.

We see that the accuracy of RigL is consistently matched, or even surpassed, when the number of subset samples is equal to the number of active connections, that is, $\gamma = 1$. This indicates that GSE is more effective at exploring the network connections than SET or RigL. We observe the same overall trend across all distributions, showing that the success of our method is not reliant on the specific characteristics of the sampling distribution. Moreover, the accuracy quickly plateaus as $\gamma > 1$, validating that it is sufficient for the subset size to be on the same order as the active set size.

Interestingly, uniform sampling demonstrates strong performance while we hypothesized that the biased distributions would be beneficial for training. We conjecture that this is due to an effective collaboration between uniform sampling and the greedy selection of the largest gradient magnitudes. We expect that as the subset size of GSE increases, the accuracy eventually drops back to the level of RigL because the largest gradient magnitude selection then becomes the dominant criteria, making the method more similar to RigL. To this end, we experimented with increasing the number of samples in the last column of Figure 2, note the logarithmic x-axis. The observations align precisely with our expectations. Since uniform sampling is also the most efficient distribution, we use it as our method in the following experiments.

### 4.3 COMPARISON WITH RELATED WORK

We compare our method to a broad range of sparsification methods in Table 1, the best accuracy among the sparse training methods is bolded. The Static method denotes training a static random sparse model. The other baselines are Lottery (Frankle & Carbin, 2018), Gradual (Zhu & Gupta, 2017), SNIP (Lee et al., 2018), GraSP (Wang et al., 2019), SynFlow (Tanaka et al., 2020), SET (Mocanu et al., 2018), RigL (Evci et al., 2020), and Top-KAST (Jayakumar et al., 2020). Our results use GSE with uniform sampling and $\gamma = 1$.

Table 1: Accuracy comparison with related work on CIFAR datasets.

| Dataset | CIFAR-10 | | | | | | | | | CIFAR-100 | | | | | | | | |
|---|---|---|---|---|---|---|---|---|---|---|---|---|---|---|---|---|---|---|
| Model | ResNet-56 | | | ViT | | | VGG-16 | | | ResNet-56 | | | ViT | | | VGG-16 | | |
| Dense | 92.7 | | | 85.0 | | | 89.0 | | | 70.1 | | | 58.8 | | | 62.3 | | |
| Sparsity | 90% | 95% | 98% | 90% | 95% | 98% | 90% | 95% | 98% | 90% | 95% | 98% | 90% | 95% | 98% | 90% | 95% | 98% |
| Lottery | 89.8 | 87.8 | 84.6 | 85.4 | 85.1 | 72.3 | 88.2 | 86.7 | 83.3 | 63.6 | 58.6 | 50.4 | 57.2 | 57.0 | 42.6 | 55.9 | 49.7 | 37.1 |
| Gradual | 91.3 | 89.9 | 88.0 | 83.5 | 79.3 | 72.2 | 89.1 | 88.2 | 85.8 | 67.2 | 64.4 | 57.1 | 55.6 | 53.3 | 43.6 | 60.1 | 57.0 | 44.5 |
| Static | 90.1 | 88.3 | 84.4 | 79.1 | 75.3 | 68.7 | 87.1 | 84.5 | 79.0 | 63.5 | 55.4 | 36.6 | 53.9 | 48.0 | 38.0 | 54.1 | 44.2 | 27.8 |
| SNIP | 89.6 | 87.8 | 82.9 | 75.1 | 70.4 | 63.8 | 88.3 | 86.8 | 10.0 | 62.6 | 55.5 | 41.7 | 49.7 | 44.1 | 34.6 | 52.3 | 39.1 | 1.0 |
| GraSP | 89.7 | 88.7 | 84.6 | 62.2 | 65.3 | 62.3 | 86.4 | 85.3 | 82.0 | 61.8 | 56.9 | 42.3 | 40.5 | 39.6 | 32.6 | 53.4 | 45.7 | 30.1 |
| SynFlow | 90.3 | 88.0 | 83.8 | 78.6 | 75.7 | 69.7 | 87.5 | 85.4 | 78.7 | 60.5 | 50.6 | 31.4 | 52.9 | 48.5 | 37.9 | 52.1 | 41.8 | 24.8 |
| SET | 90.2 | 88.8 | 85.5 | 79.4 | 75.2 | 68.8 | 86.8 | 84.8 | 79.9 | 65.0 | 60.5 | 49.7 | 54.3 | 48.2 | 38.7 | 54.7 | 47.9 | 32.4 |
| RigL | 90.6 | 89.5 | 86.7 | 79.9 | 76.2 | 70.6 | 88.5 | 87.1 | 83.7 | 65.7 | 62.3 | 52.8 | **54.6** | **49.0** | 40.4 | 57.0 | 51.8 | 39.9 |
| Top-KAST | 89.8 | 88.1 | 85.2 | 75.0 | 68.3 | 55.0 | 86.9 | 84.8 | 80.5 | 62.5 | 58.3 | 40.4 | 51.1 | 43.3 | 28.2 | 54.4 | 47.0 | 32.2 |
| GSE (*ours*) | **91.0** | **89.9** | **87.0** | **80.0** | **76.4** | **70.7** | **88.6** | **88.1** | **85.1** | **66.0** | **62.6** | **54.3** | 54.4 | 48.8 | **40.7** | **57.9** | **52.3** | **40.8** |

Among the sparse training methods (from Static downwards in Table 1) our method outperforms the other methods consistently over datasets, sparsities, and model architectures. While at 90% sparsity all the methods achieve comparable accuracy, at the extreme sparsity rate of 98%, differences between methods become more evident. At 98% sparsity our method outperforms all the before training sparsification methods by a significant margin, 4.7% on average. As observed by Tanaka et al. (2020), layer collapse occurs at 98% for SNIP with VGG-16 on CIFAR-10 and CIFAR-100.

Gradually pruning connections during training (Gradual) can still improve the accuracy, especially in the extreme sparsity regimes. However, this comes at the cost of training the dense model, thus requiring significantly more FLOPs than our method. This observation serves as motivation for further investigation into dynamic sparse training methods that improve on our accuracy while preserving our efficiency advantage. Notably, our method outperforms the Lottery Ticket Hypotheses on the CNN models, while LTH requires multiple rounds of training.

In Appendix H, we report the test accuracy of each method at every epoch throughout training, showing that our method trains as fast as the other sparse training methods. Additionally, in Appendix M, we provide plots of the layer-wise sparsity at the end of training for each combination of method, model, and sparsity. This information provides insight into how each method distributes the available connections over the layers of a model.

## 4.4 IMAGENET

To determine the efficacy and robustness of GSE on large models and datasets, in this third experiment, we compare the accuracy of ResNet-50 trained on ImageNet at 90% sparsity. In Table 2, the accuracy obtained with GSE is compared against the baselines mentioned in the preceding section, with the addition of DSR (Mostafa & Wang, 2019) and SNFS (Dettmers & Zettlemoyer, 2019). Since this is a common benchmark among sparse training methods, the accuracy results of the baselines were obtained from prior publications. Our experiments use the previously mentioned hyperparameters with the following exceptions: we train for 100 epochs and use learning rate warm-up over the initial 5 epochs, gradually reaching a value of 0.1; subsequently, the learning rate is reduced by a factor of 10 every 30 epochs; and we use label smoothing with a coefficient of 0.1. Our results use GSE with uniform sampling and $\gamma = 2$. The results in Table 2 align with those in the preceding section, consistently demonstrating GSE's superior performance over other sparse training methods. These results collectively reinforce that GSE is robust and able to scale to large models and datasets.

Table 2: ResNet-50 accuracy on ImageNet at 90% sparsity.

| Method | Accuracy |
|---|---|
| Dense | 76.8±0.09 |
| Lottery | 69.4±0.03 |
| Gradual | 73.9 |
| Static | 67.7±0.12 |
| SNIP | 67.2±0.12 |
| GraSP | 68.1 |
| SET | 69.6±0.23 |
| DSR | 71.6 |
| SNFS | 72.9±0.06 |
| RigL | 73.0±0.04 |
| Top-KAST | 73.0 |
| GSE (*ours*) | **73.2±0.07** |

## 4.5 MODEL SCALING

In this experiment, we investigate how the size of a model affects the accuracy when the number of active connections is kept constant. We scale the number of filters in convolution layers and units in fully-connected layers to obtain a wider model. The results, presented in Figure 3, show that training wider CNN models translates to significantly increased accuracy, 6.5% on average at 4 times wider. These findings support those from Zhu & Gupta (2017) and Kalchbrenner et al. (2018), however, our method enables a sparse model to be trained whose dense version would be too computationally intensive.

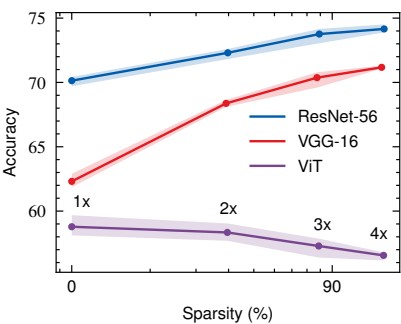

Figure 3: Increasing model size and sparsity on CIFAR-100, each line represents a fixed number of active connections.

The same trend does not hold for ViT, as its accuracy decreases with increasing scale. Appendix J reveals that the accuracy increases slower for larger ViT models, while for CNNs the accuracy increases faster as they get larger. Moreover, the accuracy of larger dense ViT models plateaus at 62.2%, as shown in Appendix K, indicating that this phenomenon is independent of our method. ViT models are known to require large amounts of training data to achieve optimal accuracy (Dosovitskiy et al., 2020), we conjecture that this contributes to the trend observed with ViT. In Appendix I, we observe that both ViT and CNN models do benefit considerably from extended training, with a 5.3% average increase in accuracy for training twice as long.

## 4.6 FLOATING-POINT OPERATIONS

Lastly, to determine the efficiency of GSE, we compare its FLOPs to those of RigL in Figure 4. The FLOPs are obtained for ResNet-50 on ImageNet using the method described by Evci et al. (2020), with $\gamma = 1$. The results show that at 99% sparsity GSE uses 11.8% fewer FLOPs than RigL, at higher sparsities GSE uses considerably fewer FLOPs. For illustration, the right plot extends the

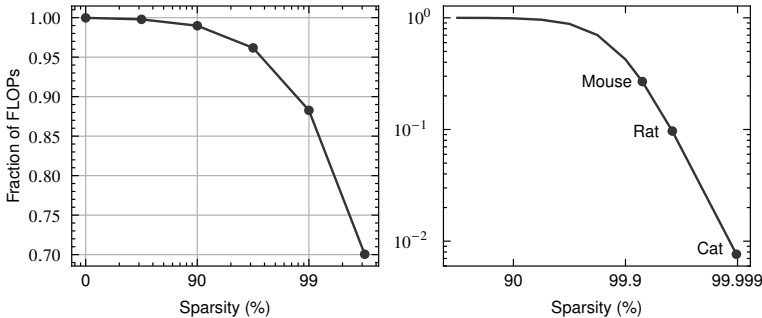

Figure 4: Fraction of FLOPs used by GSE compared to RigL for ResNet-50 on ImageNet. Left spans common ANN sparsity levels, and right illustrates the fraction of FLOPs at the sparsity of various animal brains (Ananthanarayanan et al., 2009).

sparsity to that of various animal brains which underscores the significance of our always sparse training method, as RigL uses 131 times more FLOPs to train a model with the same sparsity as the cat brain. Note that the human brain is far more sparse, thus further reducing the fraction of FLOPs.

### 4.7 LIMITATIONS

While we experiment on a range of sparsities, datasets, models, and baseline methods, we acknowledge that further exploration would enhance the robustness of our findings. Following previous work, our experiments focus on the domain of computer vision, and although we demonstrate the effectiveness of our approach within this domain, it remains to be seen how our results generalize to other domains such as natural language processing and reinforcement learning, among others. Despite this, we believe that our proposed always-sparse training algorithm, supported by its linear time complexity with respect to the model width during training and inference, represents a significant advancement. Furthermore, our method outperforms existing sparse training algorithms in terms of accuracy, as demonstrated on the CIFAR and ImageNet datasets using ResNet, VGG, and ViT models, alongside a comprehensive comparison with a range of sparsification techniques.

Secondly, our accuracy results were obtained by simulating sparse networks using binary masks. This is standard practice among sparse training methods because support for sparsity in machine learning frameworks is still in the early days, making it challenging to train models using sparse matrix formats for the weights. While we want to note that there is nothing about our method that prevents it from being accelerated on GPUs, the significant engineering required to develop an optimized GPU kernel falls outside the scope of this work. To ensure that our claims are still validated, we compare the FLOPs of our method with those of RigL in Section 4.6. Moreover, we present an unstructured sparse matrix multiplication benchmark in Appendix L, which shows that a 10x reduction in execution time is achieved at 99% sparsity, compared to dense matrix multiplication. Thus, evidently showing that real speedups can be achieved, even with unstructured sparsity.

## 5 CONCLUSION

We presented an always-sparse training algorithm which improves upon the accuracy of previous sparse training methods and has a linear time complexity with respect to the model width. This is achieved by evaluating the gradients only for a subset of randomly sampled connections when changing the connections. The three distributions for sampling the subset—uniform, an upper bound of the gradient magnitude, and an estimate of the gradient magnitude—all showed similar trends. However, uniform consistently achieves among the highest accuracy and is also the most efficient. We therefore conclude that the uniform distribution is the most appropriate for sparse training. Notably, training larger and sparser CNN models results in higher accuracy for the same number of active connections. Lastly, the number of floating-point operations of our method decrease considerably faster than those for RigL as the sparsity increases. This observation underscores the efficiency and scalability of our method, which is promising for training increasingly larger and sparser ANNs.

REPRODUCIBILITY

To ensure reproducability, the experiments used standard model architectures with any modifications stated in Section 4.1. We used the CIFAR-10/100 and ImageNet data sets with their standard train and test/validation splits. The data normalization and augmentation settings are specified in Appendices C and D, respectively. The stochastic gradient descent optimizer settings are provided in Section 4.1. The source code for the experiments is available online[1].

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

## A CONNECTION SAMPLING DISTRIBUTIONS

In this section we provide a more detailed discussion on the distributions which were considered for sampling the subset of inactive connections.

### A.1 GRADIENT MAGNITUDE UPPER BOUND (GRABO)

The gradient magnitude for a single connection $(a, b) \in \mathbb{W}$ can be expressed as follows:

$$\left|\nabla\theta_{a,b}^{[l]}\right| = \left|\sum_{i=1}^{B} h_{a,i}^{[l-1]}\delta_{b,i}^{[l]}\right| \implies \left|\nabla\theta^{[l]}\right| = \left|h^{[l-1]}\left(\delta^{[l]}\right)^{\mathsf{T}}\right| \tag{1}$$

where $h^{[l]}$ and $\delta^{[l]}$ are the activations and gradients of the loss at the output units of the $l$-th layer, respectively. When the batch size $B$ is one, the sum disappears, simplifying the gradient magnitude to a rank one matrix which can be used to sample connections efficiently, inducing a joint probability distribution that is proportional to the gradient magnitude only when the batch size is one. In practice, however, training samples come in mini batches to reduce the variance of the gradient. Therefore, we experiment with sampling the connections proportionally to the following upper bound of the gradient magnitude instead, which enables efficient sampling, even with mini batches:

$$\left|\nabla\theta^{[l]}\right| = \left|h^{[l-1]}\left(\delta^{[l]}\right)^{\mathsf{T}}\right| \leq \left(\left|h^{[l-1]}\right|\mathbf{1}\right)\left(\left|\delta^{[l]}\right|\mathbf{1}\right)^{\mathsf{T}} \tag{2}$$

The proof for Equation 2 uses the triangle inequality and is provided in Appendix B. This upper bound has the property that it becomes an equality when $B = 1$. Connections are then sampled from the probability distributions in Equation 3. The implementation of this distribution does not require any modifications to the model, and only needs a single forward and backward pass to evaluate.

$$f^{[l]} := \frac{\left|h^{[l-1]}\right|\mathbf{1}}{\mathbf{1}^{\mathsf{T}}\left|h^{[l-1]}\right|\mathbf{1}}, \quad g^{[l]} := \frac{\left|\delta^{[l]}\right|\mathbf{1}}{\mathbf{1}^{\mathsf{T}}\left|\delta^{[l]}\right|\mathbf{1}} \tag{3}$$

### A.2 GRADIENT MAGNITUDE ESTIMATE (GRAEST)

The reason we use a bound on the absolute gradient is that in general the sum of products cannot be written as the product of sums, that is, $\sum_i x_i y_i \neq (\sum_i x_i)(\sum_i y_i)$. However, it is possible to satisfy this equality in expectation. This was shown by Alon et al. (1996) who used it to estimate the second frequency moment, or the $L^2$ norm of a vector. Their work is called the AMS sketch, AMS is an initialism derived from the last names of the authors. They then showed that this also works in the more general case to estimate inner products between vectors (Alon et al., 1999). This is achieved by assigning each index of the vector a random sign, i.e. $s_i \sim \mathcal{U}\{-1, +1\}$, resulting in the following equality:

$$\sum_i x_i y_i = \mathbb{E}\left[\left(\sum_i s_i x_i\right)\left(\sum_i s_i y_i\right)\right] \tag{4}$$

We can use their finding to construct probability distributions $f^{[l]}$ and $g^{[l]}$ in Equation 5 such that their induced joint probability distribution is proportional to the gradient magnitude in expectation. This distribution can also be computed in a single forward and backward pass and requires no modifications to the model, apart from introducing a mechanism to sample the random signs $s$.

$$f^{[l]} := \frac{\left|h^{[l-1]}s\right|}{\mathbf{1}^{\mathsf{T}}\left|h^{[l-1]}s\right|}, \quad g^{[l]} := \frac{\left|\delta^{[l]}s\right|}{\mathbf{1}^{\mathsf{T}}\left|\delta^{[l]}s\right|} \tag{5}$$

## B GRADIENT MAGNITUDE UPPER BOUND PROOF

**Lemma B.1.** *The gradient magnitude of the loss with respect to the parameters of a fully-connected layer $l$ has the following upper bound:*

$$\left|\nabla\theta^{[l]}\right| \leq \left(\left|h^{[l-1]}\right|\mathbf{1}\right)\left(\left|\delta^{[l]}\right|\mathbf{1}\right)^{\mathsf{T}}$$

*where $\theta^{[l]} \in \mathbb{R}^{n^{[l-1]} \times n^{[l]}}$, $\boldsymbol{h}^{[l]} \in \mathbb{R}^{n^{[l]} \times B}$, and $\boldsymbol{\delta}^{[l]} \in \mathbb{R}^{n^{[l]} \times B}$ are the weight matrix, activations and gradients of the loss at the output units of the l-th layer, respectively. The number of units in the l-th layer is denoted by $n^{[l]} \in \mathbb{N}^+$, and $B \in \mathbb{N}^+$ is the batch size.*

*Proof.* The proof starts with the definition of the gradient of the fully-connected layer, which is calculated during back propagation as the activations of the previous layer multiplied with the back propagated gradient of the output units of the layer. We then write this definition for a single connection in the weight matrix and use the triangle inequality to specify the first upper bound. In Equation 8 we rewrite the upper bound of Equation 7 as all the cross products between the batch dimensions minus all the non-identical cross terms. Note that all the non-identical cross terms are positive because of the absolute, therefore we get another upper bound by removing the non-identical cross terms which we then write in matrix notation in Equation 9.

$$\left| \nabla \theta^{[l]} \right| = \left| \boldsymbol{h}^{[l-1]} \left( \boldsymbol{\delta}^{[l]} \right)^{\mathsf{T}} \right| \tag{6}$$

$$\Longrightarrow \left| \nabla \theta_{a,b}^{[l]} \right| = \left| \sum_{i=1}^{B} \boldsymbol{h}_{a,i}^{[l-1]} \boldsymbol{\delta}_{b,i}^{[l]} \right| \le \sum_{i=1}^{B} \left| \boldsymbol{h}_{a,i}^{[l-1]} \boldsymbol{\delta}_{b,i}^{[l]} \right| \tag{7}$$

$$= \sum_{i}^{B} \left| \boldsymbol{h}_{a,i}^{[l-1]} \right| \sum_{i}^{B} \left| \boldsymbol{\delta}_{b,i}^{[l]} \right| - \sum_{r,s \ne r}^{B} \left| \boldsymbol{h}_{a,r}^{[l-1]} \boldsymbol{\delta}_{b,s}^{[l]} \right| \le \sum_{i}^{B} \left| \boldsymbol{h}_{a,i}^{[l-1]} \right| \sum_{i}^{B} \left| \boldsymbol{\delta}_{b,i}^{[l]} \right| \tag{8}$$

$$\Longrightarrow \left| \nabla \theta^{[l]} \right| \le \left( \left| \boldsymbol{h}^{[l-1]} \right| \mathbf{1} \right) \left( \left| \boldsymbol{\delta}^{[l]} \right| \mathbf{1} \right)^{\mathsf{T}} \tag{9}$$

□

## C  DATA NORMALIZATION

We normalize all the training and test data to have zero mean and a standard deviation of one. The dataset statistics that we used are specified below, where $\mu$ is the mean and $\sigma$ the standard deviation. The values correspond to the red, green, and blue color channels of the image, respectively.

**CIFAR-10**

$$\mu = (0.4914, 0.4822, 0.4465), \quad \sigma = (0.2470, 0.2435, 0.2616)$$

**CIFAR-100**

$$\mu = (0.5071, 0.4865, 0.4409), \quad \sigma = (0.2673, 0.2564, 0.2762)$$

**ImageNet**

$$\mu = (0.485, 0.456, 0.406), \quad \sigma = (0.229, 0.224, 0.225)$$

## D  DATA AUGMENTATION

We applied standard data augmentation as part of our training data pipeline. The specific augmentation techniques are specified below. We apply the data normalization described in Appendix C after the augmentation transformations.

**CIFAR-10**  We pad the image with 4 black pixels on all sides and randomly crop a square of 32 by 32 pixels. We then perform a random horizontal flip of the crop with a probability of 0.5.

**CIFAR-100**  We pad the image with 4 black pixels on all sides and randomly crop a square of 32 by 32 pixels. We then perform a random horizontal flip of the crop with a probability of 0.5.

**ImageNet** We randomly crop a square of 224 by 224 pixels. We then perform a random horizontal flip of the crop with a probability of 0.5. For the test set, we resize the images to be 256 pixels followed by a square center crop of 224 pixels.

# E    CONVOLUTIONAL LAYER SUBSET SAMPLING

In this section we describe how our growing and pruning algorithm can be applied to convolutional layers. While a weight in a fully-connected layer is specified by the input and output units that it connects, a weight in a 2D convolutional layer is specified by the input channel, its *x* and *y* coordinate on the filter, and the output channel. It thus seems that it would require four discrete probability distributions to sample a weight of a 2D convolutional layer, instead of two for fully-connected layers. However, one can alternatively view a convolutional layer as applying a fully-connected layer on multiple subsections (patches) of the input, flattening the input channels and filters makes the weights of a convolutional layer identical to a fully-connected layer. The multiple subsections of the input can then simply be treated as additional batches. With this realization all the methods that we explain throughout the paper for fully-connected layers directly apply to convolutional layers.

# F    SPARSE GRAPH INITIALIZATION COMPARISON

In Figure 5, we plot the accuracy obtained by assigning the sparsity uniformly over all the layers compared to using the Erdős–Rényi initialization for SET, RigL, and our method. Note that Ours Uniform (Erdős–Rényi) denotes uniform sampling of the subset connections and the Erdős–Rényi initialization. The Erdős–Rényi initialization obtains better accuracy while ensuring that the number of active connections per layer is proportional to the layer width.

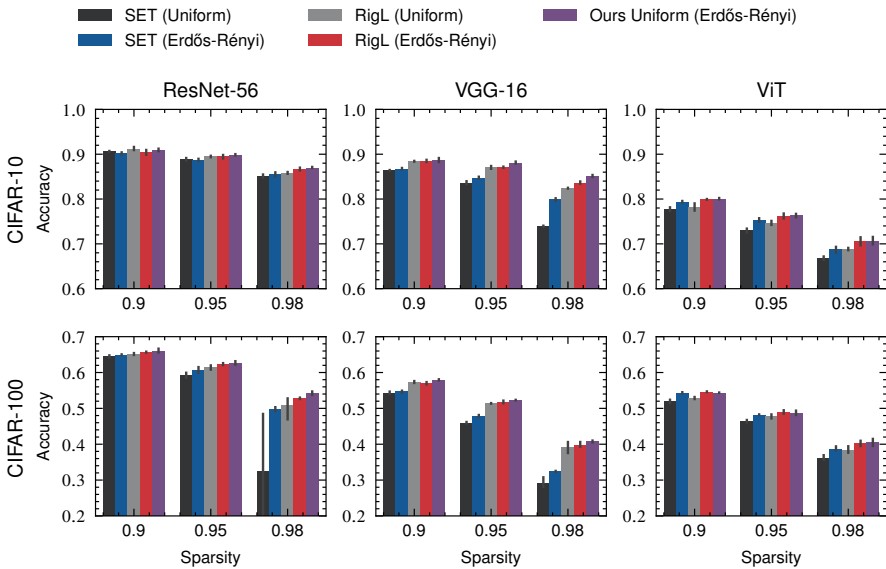

Figure 5: Accuracy comparison of uniform and Erdős–Rényi sparsity assignment.

## G  PSEUDOCODE

We provide pseudocode for our complete dynamic sparse training process in the case of the GraBo or GraEst distributions in Algorithm 2. The uniform distribution is simpler as it does not require any aggregation of activity. For simplification, the pseudocode assumes that the prune-grow step only uses a single batch of training samples, allowing the aggregation for the probability distributions f and g to be computed in the same pass as the generation of the various connection sets. This is also the scenario we tested with in all our experiments.

---

**Algorithm 2** Efficient periodic growing and pruning of connections

---

**Input:** Network $f_\theta$, dataset $\mathcal{D}$, loss function $\mathcal{L}$, prune-grow schedule $T, T_{\text{end}}, \alpha$, number of active connections multiplier $\epsilon$, and subset sampling factor $\gamma$.

1: **for** each layer $l$ **do**
2:   $\mathbb{A} \leftarrow$ sample_graph($n^{[l-1]}, n^{[l]}; \epsilon$)          ▷ Erdős–Rényi random bipartite graph
3:   $\theta \leftarrow$ sample_weights($\mathbb{A}$)
4: **end for**
5: **for** each training step $t$ **do**
6:   $\boldsymbol{x}, \boldsymbol{y} \sim \mathcal{D}$                    ▷ Sample a mini batch
7:   **if** $t \bmod T = 0$ **and** $t \leq T_{\text{end}}$ **then**
8:     $\boldsymbol{h} \leftarrow \boldsymbol{x}$
9:     **for** each layer $l$ **do**
10:       $\boldsymbol{a}^{[l]} \leftarrow$ aggregate($\boldsymbol{h}$)         ▷ Aggregate is distribution specific
11:       $\boldsymbol{h} \leftarrow$ forward($\boldsymbol{h}$)           ▷ Sparse forward pass
12:     **end for**
13:     $\boldsymbol{g} \leftarrow$ grad($\mathcal{L}(\boldsymbol{h}, \boldsymbol{y})$)         ▷ Get the gradients at the outputs
14:     **for** each layer $l$ in inverse order **do**
15:       $\boldsymbol{b}^{[l]} \leftarrow$ aggregate($\boldsymbol{g}$)
16:       f $\leftarrow$ normalize($\boldsymbol{a}^{[l]}$)        ▷ Create probability distribution
17:       g $\leftarrow$ normalize($\boldsymbol{b}^{[l]}$)
18:       $\mathbb{S} \leftarrow$ sample_connections(f, g, $\lceil \gamma |\mathbb{A}| \rceil$) $\setminus \mathbb{A}$    ▷ Sample the connections subset
19:       $\alpha_t \leftarrow$ cosine_decay($t; \alpha, T_{\text{end}}$)      ▷ Maximum prune fraction
20:       $k \leftarrow \min(\lceil \alpha_t |\mathbb{A}| \rceil, |\mathbb{S}|)$      ▷ Amount to prune and grow
21:       $\mathbb{G} \leftarrow$ topk($|\text{grad}(\mathbb{S})|, k$)      ▷ Get grown connections set
22:       $\mathbb{P} \leftarrow$ topk($-|\theta|, k$)        ▷ Get pruned connections set
23:       $\mathbb{A} \leftarrow (\mathbb{A} \setminus \mathbb{P}) \cup \mathbb{G}$     ▷ Construct new active connections set
24:       $\theta \leftarrow$ update_weights($\theta, \mathbb{A}, \mathbb{P}, \mathbb{G}$)
25:       $\boldsymbol{g} \leftarrow$ backward($\boldsymbol{g}$)        ▷ Sparse backward pass
26:     **end for**
27:   **else**
28:     SGD($\theta, \mathcal{L}(f_\theta(\boldsymbol{x}), \boldsymbol{y})$)        ▷ Regular SGD training
29:   **end if**
30: **end for**

---

## H    TRAINING PROGRESS

In Figure 6, we provide plots for the test accuracy during training for each baseline method and our method. It can be seen that our method consistently achieves among the highest test accuracy throughout training compared to the sparse training methods. The training progress of Gradual stands out because its accuracy increases the most at first, this is because it starts out as a dense model while it gradually increases the model sparsity, as a result, the accuracy goes down towards the second learning rate drop when the target sparsity is reached.

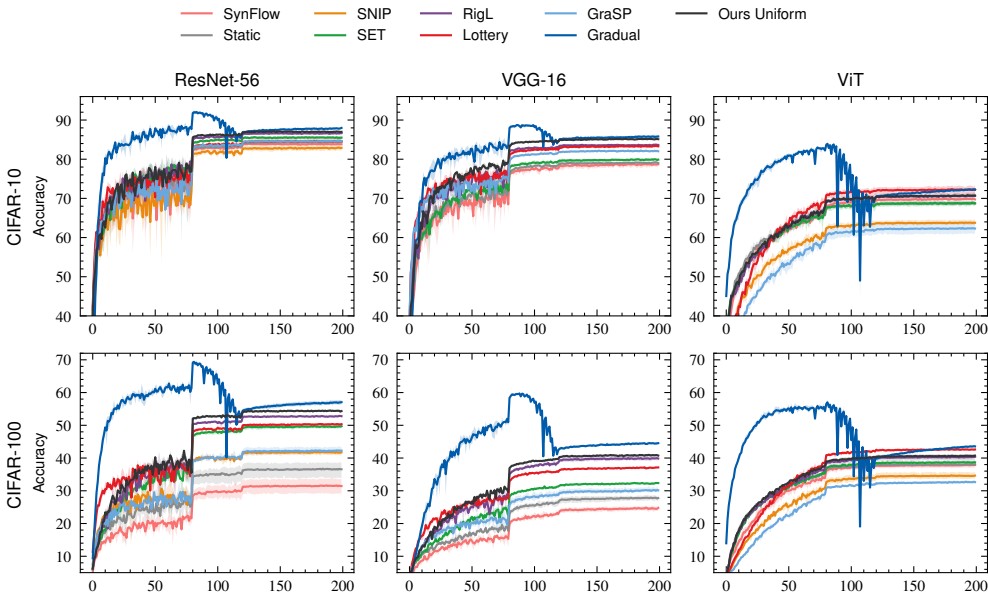

Figure 6: Training progress in test accuracy compared to related work at 98% sparsity.

## I    EXTENDED TRAINING

The following extended training results are for CIFAR-100 at 98% sparsity with 1x, 1.5x, and 2x the number of training epochs. The learning rate drops and the final prune-grow step are also scaled by the same amount, all other hyperparameters stay the same. It is clear that all models benefit significantly from extended training. The ViT model in particular gains 7% in accuracy by training for twice as long.

Table 3: Accuracy of extended training on CIFAR-100 at 98% sparsity

| Epochs | 200 (1x) | 300 (1.5x) | 400 (2x) |
|---|---|---|---|
| ResNet-56 | 54.3±0.5 | 56.4±0.3 | 57.9±0.1 |
| VGG-16 | 40.8±0.3 | 44.1±0.4 | 45.9±0.2 |
| ViT | 40.7±1.0 | 45.3±1.1 | 47.8±0.5 |

## J    MODEL SCALING TRAINING PROGRESS

In Figure 7, we provide plots for the test and training accuracy during training for our method on CIFAR-100 while increasing the width of the model by 1x, 2x, 3x, and 4x. The number of active connections is kept constant between all experiments, see Section 4.5 for details. In the CNN models, ResNet-56 and VGG-16, the increase in width causes the models to train significantly faster and obtain higher accuracy. For the vision transformer (ViT) model, the trend is reversed, wider models train slower and obtain lower accuracy.

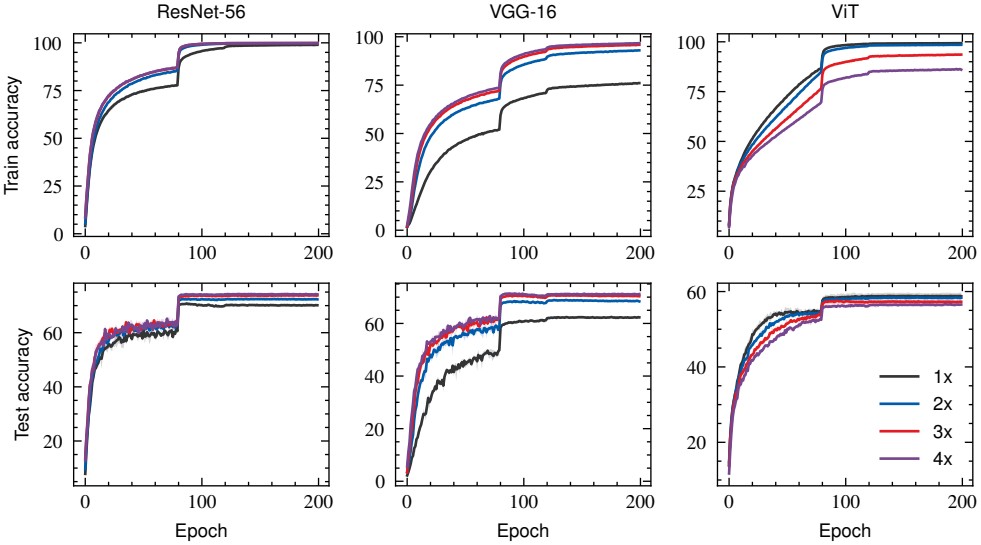

Figure 7: Training progress in train and test accuracy for various model widths on CIFAR-100.

## K    SCALING DENSE VIT

To further investigate the behavior of ViT, which decreases in accuracy when using larger models with the same number of activate parameters, as discussed in Section 4.5 and Appendix J. In Table 4, we report the accuracy of ViT for increasing model width but kept dense, thus increasing the active connections count. We see the accuracy quickly plateaus at 62.2%, gaining only 2% accuracy while even the sparse CNNs gained 6.5% on average at 4 times wider. This indicates that the decreasing accuracy of ViT reported in Section 4.5 is not due to our sparse training method as even the dense model does not benefit much from increased scale.

Table 4: Accuracy of scaling dense ViT on CIFAR-100

| Model width | 1x | 2x | 3x | 4x |
|---|---|---|---|---|
| Accuracy | 60.3±0.6 | 62.5±0.5 | 62.2±0.4 | 62.2±0.1 |

## L  SPARSE MATRIX MULTIPLY BENCHMARK

In Figure 8, we show the execution time of the sparse times dense matrix-matrix multiply on two execution devices: Intel Xeon Gold 6148 CPU at 2.40GHz and NVIDIA Tesla V100 16GB. The sparse matrix is a square matrix of size units times units, representing an unstructured sparse random weight matrix. The dense matrix is of size units times 128 and represents a batch of layer inputs. We compare three storage formats, namely: dense, where all the zeros are explicitly represented; Coordinate format (COO), which represents only the nonzero elements and their coordinates; and Compressed Sparse Row (CSR), which is similar to COO but compresses the row indices. Each data point in Figure 8 is the average of 10 measurements.

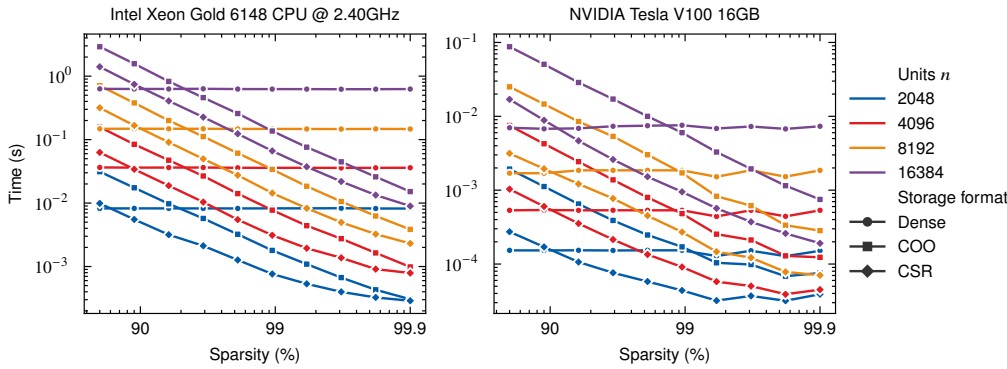

Figure 8: Execution time of sparse times dense matrix-matrix multiply.

The results show that the CSR format generally outperforms the COO format. Moreover, the execution time of the CSR format improves upon the dense execution time starting around 90% sparsity. At 99% sparsity, the CSR format is up to an order of magnitude faster than the dense format. These results are similar between the two execution devices, although the NVIDIA Tesla V100 16GB is up to two orders of magnitude faster than the Intel Xeon Gold 6148 CPU at 2.40GHz.

## M  LAYER-WISE SPARSITY

In Figure 9, we show the sparsity of each layer, in terms of compression, obtained by each sparsification methods at the end of training. In the plots the CIFAR-10 and CIFAR-100 results are averaged. These plots can be used to gain insight into how each sparsification method distributes the active connections over the models.

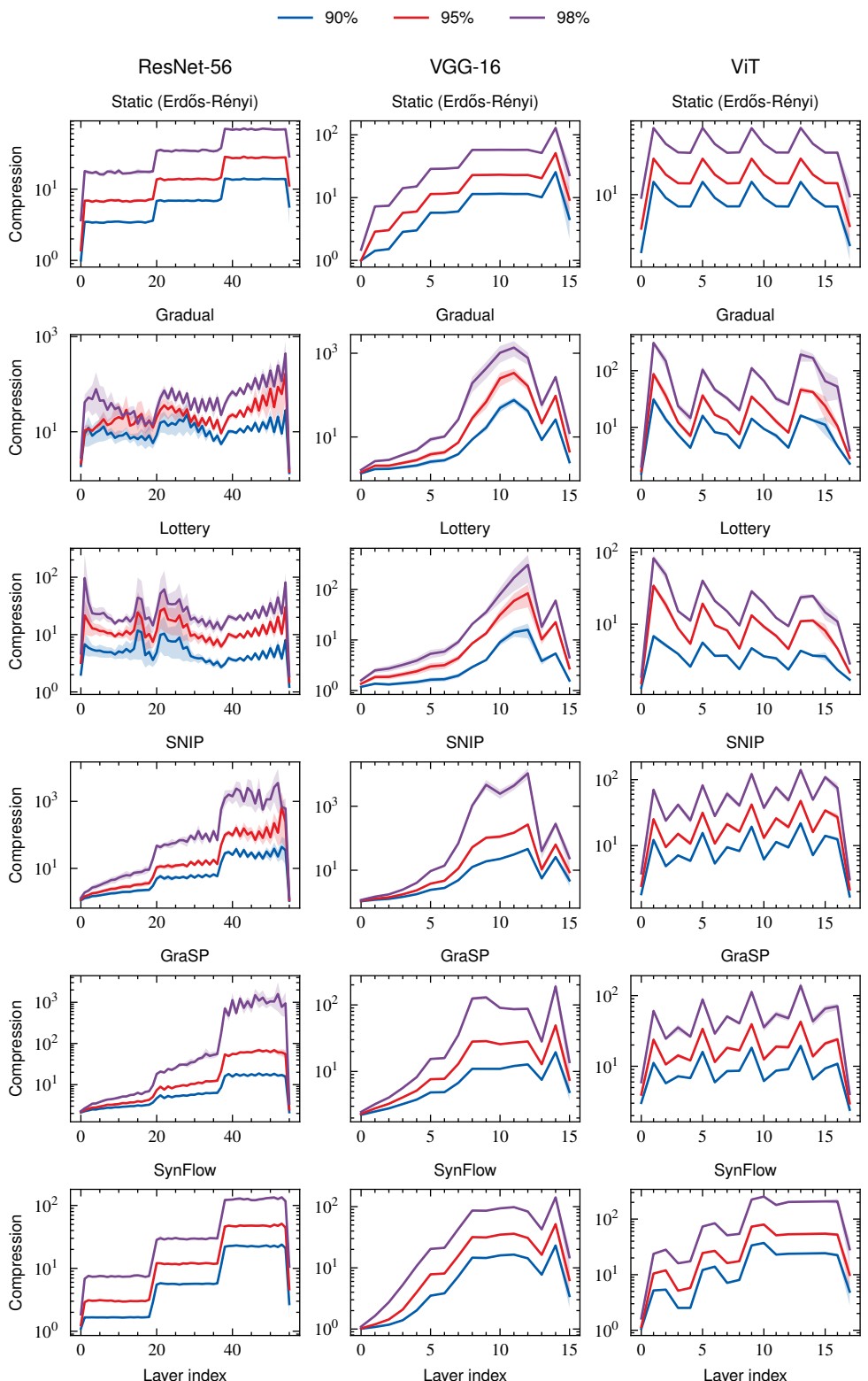

Figure 9: (Part 1) Layer-wise sparsity from each sparsification method at the end of training. Showing averaged results on the CIFAR-10/100 datasets.

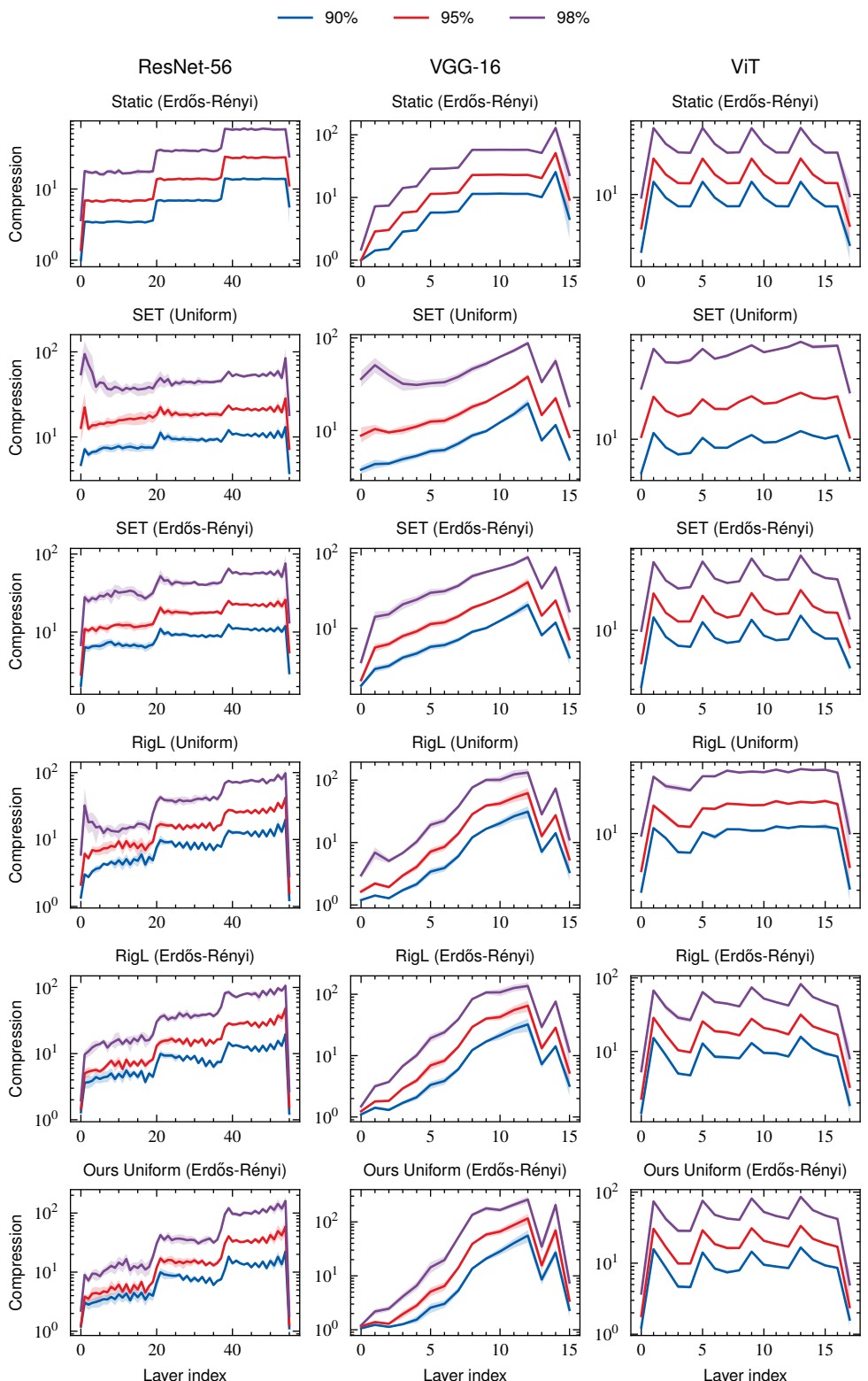

Figure 9: (Part 2) Layer-wise sparsity from each sparsification method at the end of training. Showing averaged results on the CIFAR-10/100 datasets.

