# OpenReview forum: "Always-Sparse Training with Guided Stochastic Exploration"
_ICLR.cc/2024/Conference — Submitted to ICLR 2024_

### Official Review · Reviewer_bMwd · 2023-10-15

**Soundness:** 3 good
**Presentation:** 3 good
**Contribution:** 1 poor
**Rating:** 3
**Confidence:** 4

**Summary:**

The paper presents an approach to apply a grow and prune approach to train an "always sparse" neural network. The paper presents how guided stochastic exploration can be used in a grow and prune approach to train an always sparse neural network with some amount of empirical performance. The paper presents an approach to do guided stochastic exploration using subset sampling, wrapped in the overall "cosine decay" strategy of decision making. The cosine decay strategy is well known for having some form of, "explore-exploit," baked in. Empirical validation and comparisons to related work in the field is presented. Some vague connections to actual neural networks in human brains is hinted at.

**Strengths:**

The authors have presented a well written and understandable argument for their paper to be accepted.

**Weaknesses:**

The paper delivers no actual empirical results useful in practice. As deep learning is a practical and engineering field, it is not really known what is the value of showing improvements in big-O format, when actual implementation of these approaches are in super huge networks such as ChatGPT, GPT4, Diffusion Models, and many other productionized models. There are open source minified models of the above as well that can be run on a desktop or a laptop. The reviewer believes that there are no practical benefits of this approach in that setting either.

If the paper should be acceptable as a theoretical work due to the potential of its' contributions being useful in the future, then the authors have not shown sufficient novelty compared to the many other related works that are cited within the work. The technical tools used to deliver improvements are rather well known with obvious applications to this problem. If the paper is meant to be in a theoretical setting, what model of computation should be used to analyze its claims?

The paper does not consider over wall-clock time analysis in any of its thinking. If the procedure itself of guided stochastic exploration takes longer than the theoretical speedup in FLOPs gained by the sparsifying, then, then any benefits of the proposed approach is moot.

The paper considers sparsifying weight connections, it is well known that this cannot, in practice, deliver any actual improvement on massively parallel SIMD architectures used in deep learning today. Sparse general matrix-matrix multiplies are known to be, more or less, impossible to speed up in SIMD setting where the actual observed wall-clock time speedup is anywhere close to the theoretical speedup indicated by FLOP reduction. There are several peer reviewed works exploring this problem.

The connection between artificial neural networks and actual neural networks in the brains of living creatures is not well thought out, I will cover this in more detail in the "Questions" section of the review.

**Questions:**

What do the authors believe is a reasonable path to move this paper towards some form of acceptance given the above concerns?

If the purpose of this paper is to reason or advance the field of "biologically plausible neural networks" that may have applications to neuroscience, could the authors cite why this paper would be valuable to the neuroscience community? Could the authors make that argument iron-tight and clear before it is considered under the peer review setting?

If the purpose of this paper is to connect the field of Artificial Neural Networks with "biological neural networks" in brains of living creatures, could the author expound on this greatly? One place to start would be how actual neurons in actual brains having spiking activity called spike trains, and don't behave much like artificial neural networks at all. Spiking neural networks are also a good place to start on this path.

If it is the case that this paper is an exploratory, prototyping work for some other future work based on the findings of this research work, do the authors believe that it is sufficient to meet the bar for peer review?

---

> ### Author Response · Authors · 2023-11-17
>
> Thank you for taking the time to review our paper. We respectfully disagree with your assessment. There is an established line of peer-reviewed work that aims to deliver the highest accuracy for a given FLOPs budget by training sparse models. In this line of work, our method improves upon both the FLOPs and the accuracy, thus yielding a strict improvement over previous methods. Our time complexity analysis further supports this claim.
>
> The vast majority of the DST literature uses FLOPs to compare the efficiency of various methods. This is because support for efficient sparse implementations of common neural network operations is currently missing in machine learning frameworks, as noted in our limitations section, making wall-time experiments challenging. However, we also explicitly highlight that real speedups, even with unstructured sparsity, can be obtained at high levels of sparsity as demonstrated in Appendix L.
>
> We want to stress that we make no claims throughout the paper about any connections between our method and biologically plausible neural networks. We mention the sparsity level of some biological brains to indicate that such high levels of sparsity exist. However, even at the levels of sparsity that are common within the sparse ANN literature, our method provides improvements in FLOPs as shown in Figure 4.
>
> While we disagree with your assessment, we still greatly appreciate that you took the time to review our work and we hope that our explanation helps to clarify our approach.

---

> ### Comment · Reviewer_bMwd · 2023-11-17
> **Followup comment**
>
> My specific comments
>
> - If the purpose of this paper is to reason or advance the field of "biologically plausible neural networks" that may have applications to neuroscience, could the authors cite why this paper would be valuable to the neuroscience community? Could the authors make that argument iron-tight and clear before it is considered under the peer review setting?
>
> - If the purpose of this paper is to connect the field of Artificial Neural Networks with "biological neural networks" in brains of living creatures, could the author expound on this greatly? One place to start would be how actual neurons in actual brains having spiking activity called spike trains, and don't behave much like artificial neural networks at all. Spiking neural networks are also a good place to start on this path.
>
> Was to broadly evaluate this paper within the context of the intersectional field of ANNs and neuroscience, which from what I understand is a well established field. It could always be the case that developments in ANN pruning can improve our understanding of neuroscience and vice versa.
>
> Given that the authors say that this is not the case. After looking at Appendix L, here are my followup thoughts:
>
> - This is very much tacked on, where the authors move the actual difficult question to "outside the scope of this work."
> - The reviewer believes that this should be the scope of the work, being able to show practical improvements.
> - The authors themselves say it requires 90%+ or higher sparsity to achieve any speedup at all. Do the authors believe this tradeoff is worth it?
> - Note that sparsifying matrices naturally degrades the performance of the neural network. So following this line of reasoning, I would have to sparsify up to 95%? 99%? in order to achieve a reasonable speedup. How much would the performance degrade by that point?
> - It's very difficult to reason about the plots in Appendix L without also seeing, in the same plot, the performance degradation for that amount of speedup.
>
> If I have missed something, I am happy to withdraw my comment.

---

> ### Comment · Reviewer_bMwd · 2023-11-18
> **Additional comment on Sparse-sparse matrix multiply (SpGEMM) vs Sparse-dense matrix multiply**
>
> I was quite confused as to why the authors were using Sparse-dense matrix multiply, instead of SpGEMM. Then I realized that SpGEMM would show up in convolutional filter pruning, whereas Sparse-dense would show up in connection weight pruning.
>
> I am still very worried about the plots in Appendix L on evaluating this relationship. For example, in both plots, the x-axis starts at 6 ticks off from 90% sparsity. So the left-most datapoint is already at 30% sparsity.
>
> Here's the problem, from what I can see. Looking at solely the largest (purple) evaluation. If you look at the less massively parallel architecture (CPU) with much less SIMD, the point at which Sparse-dense becomes faster than dense-dense is much earlier than in the V100 architecture. On the CPU it's around 91% sparse. For the GPU it's around 97-98% sparse. Extrapolating this to more modern hardware such as H100 or A100, is the relationship going to be even worse? What about extrapolating to multi-gpu execution?

---

> ### Author Response · Authors · 2023-11-19
>
> We appreciate the reviewer's timely reply. We fully agree that there is an opportunity to establish connections between the presented work and neuroscience. With our previous comment we wanted to clarify that we do not investigate such connections explicitly in our current work, but we do see the value in what our method enables: efficiently training large models with the sparsity of biological neural networks. We are eager to discover the insights that will come from investigating these connections.
>
> A quick aside on terminology, we will use MM to denote dense-dense matrix-multiply, SpMM for sparse-dense matrix-multiply, and SpSpMM for sparse-sparse matrix-multiply.
>
> While in the limitations section we state that "the significant engineering required to develop an optimized [sparse] GPU kernel falls outside the scope of this work", this does not in any way mean that we consider it outside the scope of our work to present evidence for the practical improvements of our method, quite the contrary. We present the SpMM benchmark in Appendix L and extend the CIFAR experiments to 98% sparsity exactly because we are aware of limitations within the sparse training literature which heavily relies on FLOPs comparisons rather than showing practical speed improvements.
>
> It is important to note that the benchmark in Appendix L does not represent the limit of SpMM efficiency, it should rather be interpreted as a snapshot of the current state of SpMM in machine learning frameworks (PyTorch in particular). It is entirely plausible that advances in both hardware and software will improve the efficiency of SpMM, see for instance two recent works [1, 2]. Perhaps the reviewer was referring to the COO sparse matrix format results in Appendix L, which are indeed less efficient on the GPU, breaking even around 98% sparsity. However, using the CSR sparse matrix format, both the CPU and GPU start showing speedups from 90% sparsity. Moreover, at 99% sparsity, SpMM is around 10 times faster on both the CPU and GPU, compared to MM. Despite concerns by the reviewer, there is no reason to believe that these results will be affected by multi-GPU execution using either data or model parallelism.
>
> Even with the current state of SpMM, we can thus conclude that real speedups are possible. If we consider the reported VGG-16 results on CIFAR-10, at 98% sparsity a 3.9% drop in accuracy is obtained compared to the dense model. However, 98% sparsity would result in close to 10 times faster training. We believe that there certainly are scenarios where trading 3.9% accuracy for an almost 10 times speedup is entirely reasonable.
>
> We greatly appreciate the reviewer's willingness to engage in dialogue about the presented work, and the opportunity to underscore important considerations regarding our work.
>
> [1] Huang, Guyue, et al. "Ge-spmm: General-purpose sparse matrix-matrix multiplication on gpus for graph neural networks." SC20: International Conference for High Performance Computing, Networking, Storage and Analysis. IEEE, 2020.
>
> [2] Gerogiannis, Gerasimos, et al. "SPADE: A Flexible and Scalable Accelerator for SPMM and SDDMM." Proceedings of the 50th Annual International Symposium on Computer Architecture. 2023.

---

> > ### Comment · Reviewer_bMwd · 2023-11-20
> > **Followup response**
> >
> > Thanks, the author is indeed correct that I misread the wrong line in the plots. After rereading it, I agree that in both cases, parity is achieved at 90% sparsity.
> >
> > I also thank the authors for the two followup citations. I do know that there are always recent works in this area, however I don't keep perfectly up to date on every work. I do think, approximately, that this is a difficult area to make progress on with much of the improvements being upper bounded by some constant factor with diminishing returns.
> >
> > I still think, this is a subtle question without a straightforward answer. Although it is easy to claim that such a trend would hold to newer hardware, and multi-gpu execution, it would be nicer to see this data first. For example, such a "nice" pattern may not hold in complex network architectures, whereas the authors have provided synthetic experiments only. It could be the case that this pattern does hold in multi-gpu execution as often a minibatch is split over many gpus, and synchronization means averaging the gradients, which is perfectly parallelizable. On the other hand, as we grow to larger hardware, Sparse-dense matrix multiplies (where parallelization is over the minibatch) can only be used as long as the minibatch continues to grow. However, from what I know, there is some cost to parallelizing matrix multiplies if the minibatch doesn't grow, but the network sizes continue to grow. So there may be some more difficulty here that the authors haven't considered. The same is true in multi-gpu execution where (for some reason) the minibatch doesn't grow but the size of the networks does. In essence, the crux of my argument is that it is unclear what the true constants are in terms of speedup vs. performance degradation with the current presentation. And thus, difficult to evaluate for sure.
> >
> > With regards to connection between ANNs and biological neural networks. I am not the best reviewer to evaluate the paper along that dimension, so I defer to the other reviewers.

---

> > > ### Author Response · Authors · 2023-11-22
> > >
> > > We appreciate the reviewer's insightful comments regarding the scalability of sparse-dense matrix multiplication in the context of large-scale multi-gpu training. We agree that this is a complex and nuanced topic that warrants further exploration beyond the scope of our current manuscript. As the reviewer rightly points out, the performance of sparse-dense matrix multiplication can vary depending on factors such as network architecture, hardware configuration, and minibatch size.
> > >
> > > In light of these considerations, we propose that a detailed discussion be deferred to a future manuscript specifically focused on this topic. This will allow for the in-depth analysis that the reviewer aptly suggests, while maintaining the current work's focus on the core contribution of our always-sparse training algorithm. We welcome the reviewer's opinion on whether this approach is appropriate. We believe that a dedicated manuscript would provide a valuable contribution to the field of large-scale deep learning and would complement our current work. Additionally, we will acknowledge the reviewer's insightful comments and the intricacies involved, ensuring that their valuable feedback is reflected in our current work. We really appreciate the reviewer's constructive input, which has undoubtedly strengthened the presentation of our research.

---

### Official Review · Reviewer_XMAg · 2023-10-26

**Soundness:** 3 good
**Presentation:** 2 fair
**Contribution:** 1 poor
**Rating:** 3
**Confidence:** 4

**Summary:**

This work proposes a method of always sparse training that doesn’t require materialization of full dense weight and gradient matrices at any training step. Sparsity distribution for each layer is induced based on the activation and gradients of a given layer. The approach is validated on a couple of vision tasks and architectures.

**Strengths:**

The approach for search of pruned/active connections as well as sparsity distributions make sense. The method manages to match or even outperform the baseline RigL method in a couple of scenarios.

**Weaknesses:**

The contribution of the work seems to be largely incremental. Introduced **GSE** is based on RigL and only augments it with a connection sampling strategy and use of sparse gradient for determination of the new active connections.

While the value proposition of the method is the training of large-scale models that cannot fit onto consumer or high-end GPU, experiments in this work are done at small scale, amenable to researchers and practitioners with limited compute power. The potential advantage of the method could be in reduction of the FLOPs for backward pass, but the baseline RigL requires materialization of dense gradient only on update steps, done typically every 100–1000 steps. Thus, the computational savings are minor.  At this scale, a method allowing for higher exploration of sparsity masks is likely to get better performance [1], [2] for a given target sparsity and training time.

Some more recent works with always sparse training managed to significantly outperform RigL. Powerpropagation+Top-KAST [3] and ITOP-RigL [4] have noticeably better
numbers for 90% sparsity on ResNet-50 while adopting the same training procedure.

*Minor*.  Notation for $f^{[l]}$, $g^{[l]}$ is confusing. $h_l$, $g_l$ - denote a scalar value - norm of the weights / gradients for a given layer, where $1$ is a vector of all ones with length - `(num_layers,)`. And there is expression in the denominator $1^T |h/g^{(l-1)}| 1$ that would imply, that
$h^{(l)}$ is a matrix of size `(num_layers, num_layers)`. I think it would be better to write sum over $l$ in the denominator.

---
[1] Peste, Alexandra, et al. "Ac/dc: Alternating compressed/decompressed training of deep neural networks." Advances in neural information processing systems 34 (2021): 8557-8570.

[2] Kusupati, Aditya, et al. "Soft threshold weight reparameterization for learnable sparsity." International Conference on Machine Learning. PMLR, 2020.

[3] Schwarz, Jonathan, et al. "Powerpropagation: A sparsity inducing weight reparameterisation." Advances in neural information processing systems 34 (2021): 28889-28903.

[4] Liu, Shiwei, et al. "Do we actually need dense over-parameterization? in-time over-parameterization in sparse training." International Conference on Machine Learning. PMLR, 2021.

**Questions:**

How is the norm of activations / gradient computed? Does one sum all elements in a tensor and take the absolute value, or computes L1, L2 norm. The first approach would yield zero in expectation for symmetric distributions and may not reflect presence of large elements in activation/gradient tensor, whereas the latter two yield the same value after multiplication by sign.

---

> ### Author Response · Authors · 2023-11-17
>
> We appreciate the reviewer's insightful comments and have carefully considered the concerns raised. We acknowledge that many of the DST methods, including ours, are designed to benefit the training of large models at high levels of sparsity. However, performing those experiments is extremely costly and would severely limit the reproducibility of the results. It is important to emphasize that the results on FLOPs presented in Figure 4 already take into account that RigL computes the dense gradients periodically. Despite this, our method is able to reduce the overall training FLOPs, especially at high levels of sparsity.
>
> Thank you for bringing the interesting Powerpropagation work to our attention. Both In-Time Over-Parameterization (ITOP) and Powerpropagation require a DST method and are therefore orthogonal improvements. They could thus be combined with our method without undermining our core contribution. ITOP performs an extensive hyperparameter search while Powerpropagation proposes to raise the weights (element-wise) to a constant positive power, which they show results in a rich-get-richer dynamic of the weights such that the large magnitude weights get larger while the small magnitude weights get smaller. They state that this could be useful in combination with sparse training methods as they would benefit from a regularizer which promotes weight sparsity.
>
> Let us clarify the notation used for the sampling distributions. The vectors $f^{[l]}$ and $g^{[l]}$ contain the probability that each input/output is sampled in the $l$-th layer. Let's say layer $l$ has $n$ inputs and $m$ outputs, then $f^{[l]} \in R_{+}^{n}$ and $g^{[l]} \in R_{+}^{m}$, that is, they are non-negative real vectors. Sampling one index from $f^{[l]}$ and one from $g^{[l]}$ fully specifies a weight. Each weight is thus sampled by the probability specified by the outer product $f^{[l]} \otimes g^{[l]}$, which is never explicitly computed. The computation of $f^{[l]}$ and $g^{[l]}$ is performed with information local to the $l$-th layer. With GraBo, for example, the (element-wise) absolute of the activations $|h^{[l-1]}| \in R_{+}^{n \times b}$, where $b$ is the batch size, are summed over the batch size in the numerator and summed over both the batch size and the inputs in the denominator to ensure that $f^{[l]}$ is a proper probability distribution. The same goes for the definition of $g^{[l]}$ but it uses the (element-wise) absolute of the gradients at the outputs $|\delta^{[l]}| \in R_{+}^{m \times b}$ instead. We will adjust the explanation of our method in the paper accordingly to improve the clarity.
>
> Once again, we appreciate your feedback and suggestions, and we hope that our explanation helps to clarify our approach.

---

### Official Review · Reviewer_hGq2 · 2023-10-29

**Soundness:** 1 poor
**Presentation:** 2 fair
**Contribution:** 1 poor
**Rating:** 3
**Confidence:** 5

**Summary:**

This paper revises the growth criteria within sparse training algorithms employing a prune-and-grow approach. Their primary objective is to prevent the regrowth of pruned weights by confining candidate weights exclusively to those that were not just pruned. Moreover, they impose additional restrictions by selecting candidate weights as a subset through random sampling based on specific distributions. The authors proceed to perform experiments across various datasets and with different backbone models, leading to some improvements in the results.

**Strengths:**

The experiments encompass a variety of datasets, such as CIFAR-10, CIFAR-100, and ImageNet, while utilizing different backbone architectures like ResNet, ViT, and VGG. Furthermore, they conduct extensive comparisons with numerous baseline methods.

**Weaknesses:**

1. Lack of novelty

1.1 The novel insights presented in this paper are quite limited. Essentially, the proposed approach involves initially selecting a candidate subset from the weights that were not pruned, followed by weight selection based on existing criteria. Importantly, the same pruning and growth criteria are retained from previous methods.

1.2 Regarding the distribution methods employed for sampling, the use of gradient-based distributions appears to contradict the proposed efficiency enhancement. This is because these distributions also require the computation of the full gradient, resulting in a complexity that is identical to RigL [1]. Furthermore, these distributions do not yield any performance improvements over RigL [1], indicating their relatively inconsequential role in this study.

2. Marginal improvements

The improvements observed in the results are rather modest, typically falling within the range of 0.1% to 0.5%, thus demonstrating limited practical significance.

[1] Utku Evci, Trevor Gale, Jacob Menick, Pablo Samuel Castro, and Erich Elsen. Rigging the lottery: Making all tickets winners. In International Conference on Machine Learning, pp. 2943–2952. PMLR, 2020.

**Questions:**

If the size of the candidate weights were to expand to encompass all weights, the performance should theoretically match that of RigL. However, Figure 2 in the current presentation only illustrates the scenario where the maximum value of $\gamma$ is set to 2. It would be insightful to have a comprehensive plot that covers a broader range of $\gamma$ values. The existing experimental results seem to suggest that a $\gamma$ value of 1 outperforms RigL (where $\gamma$ approaches infinity). It would be valuable to discern at what point the performance begins to decline for specific $\gamma$ values and provide an explanatory analysis in this regard.

---

> ### Author Response · Authors · 2023-11-17
>
> Thank you for your valuable feedback on our paper. Regarding your inquiry about the results in Figure 2, the details you are looking for can be found in the 4th column of the figure, where the subset sampling factor is extended up to 100. Interestingly, this column shows exactly the specific behavior you mentioned. In Section 4.2, we explicitly state that "we expect that as the subset size of GSE increases, the accuracy eventually drops back to the level of RigL because the largest gradient magnitude selection then becomes the dominant criteria, making the method more similar to RigL. To this end, we experimented with increasing the number of samples in the last column of Figure 2, note the logarithmic x-axis. The observations align precisely with our expectations." We will update the caption of Figure 2 to explicitly highlight the difference of the x-axis.
>
> While our method adopts commonly used weight magnitude pruning and gradient magnitude growing criteria from the existing literature, our novelty stems from a strategic use of these common criteria. Instead of naively applying them over all possible options, we propose a method that applies the criteria on an informed subset of the weights which we show improves both efficiency and accuracy. Applying common criteria does not diminish the innovation, rather, it grounds our work on theoretically-motivated and peer-reviewed research.
>
> The differences in accuracy are more prevalent at higher levels of sparsity, at 98% sparsity the proposed method outperforms the others by up to 2%. We believe therefore that our method constitutes more than just a marginal improvement, especially when considering that our method is also strictly more efficient.
>
> We acknowledge your concern regarding the sampling distributions and wish to emphasize a crucial aspect central to our paper's premise and the proposed method. Contrary to the impression raised, the design of the sampling techniques ensures that they do not necessitate the computation of gradients for all weights. The core innovation lies exactly in the complete avoidance of dense computations and materialization throughout the entire training process. Our sampling approaches exclusively require the input activations and output gradients of a layer, these are significantly more efficient to obtain than the gradients for all the weights of a layer. This efficiency is attributed to the utilization of sparse backward passes rather than the more resource-intensive dense computations.
>
> Let us clarify the sampling process. The vectors $f^{[l]}$ and $g^{[l]}$ contain the probability that each input/output is sampled in the $l$-th layer. Let's say layer $l$ has $n$ inputs and $m$ outputs, then $f^{[l]} \in R_{+}^{n}$ and $g^{[l]} \in R_{+}^{m}$, that is, they are non-negative real vectors. Sampling one index from $f^{[l]}$ and one from $g^{[l]}$ fully specifies a weight. Each weight is thus sampled by the probability specified by the outer product $f^{[l]} \otimes g^{[l]}$, which importantly is never explicitly computed. The computation of $f^{[l]}$ and $g^{[l]}$ is performed with information local to the $l$-th layer. With GraBo, for example, the (element-wise) absolute of the activations $|h^{[l-1]}| \in R_{+}^{n \times b}$, where $b$ is the batch size, are summed over the batch size in the numerator and summed over both the batch size and the inputs in the denominator to ensure that $f^{[l]}$ is a proper probability distribution. The same goes for the definition of $g^{[l]}$ but it uses the (element-wise) absolute of the gradients at the outputs $|\delta^{[l]}| \in R_{+}^{m \times b}$ instead. We will adjust the explanation of our method in the paper accordingly to improve the clarity.
>
> Thank you again for your valuable feedback. We appreciate the opportunity to clarify these nuances and underscore the careful considerations made in our method's design, which improves upon both the efficiency and accuracy of previous methods.

---

### Meta-Review · Area_Chair_AgVx · 2023-12-08

**Metareview:**

This submission studies the problems of sparse NN learning with the explicit goal of efficiency improvement during training time, not just at inference. This is done through Always-Sparse Training, which avoids materialisation of dense weights or computation of dense gradients during training, unlike other methods.

The strengths of the approach are diverse experiments (hGq2), an outperformance of RiGL (XMAg), and the formulation of a well-written argument for acceptance (bMwd). Weaknesses are lack of novelty (hGq2, XMAg), modest empirical improvements (hGq2), and lack of empirical usefulness (bMwd). Unfortunately, as the rebuttal itself failed to make convincing arguments for reviewers to raise their scores, the submission in its current form does not meet the requirements for publication. While the authors argue for more engagement of the reviewers, this could have been achieved by providing a more concrete response to suggestions (e.g. a comparison with ITOP or Powerpropagation for reviewer XMAg).

**Justification For Why Not Higher Score:**

Unanimous rejection rating by all reviewers.

**Justification For Why Not Lower Score:**

N/A

---

### Decision · Program_Chairs · 2024-01-16

Reject